

# Modulation induced transport signatures in correlated electron waveguides

**Gal Shavit and Yuval Oreg**

Department of Condensed Matter Physics,
Weizmann Institute of Science, Rehovot, Israel 76100

## Abstract

Recent transport experiments in spatially modulated quasi-1D structures created on top of $LaAlO_3/SrTiO_3$ interfaces have revealed some interesting features, including phenomena conspicuously absent without the modulation. In this work, we focus on two of these remarkable features and provide theoretical analysis allowing their interpretation. The first one is the appearance of two-terminal conductance plateaus at rational fractions of $e^2/h$. We explain how this phenomenon, previously believed to be possible only in systems with strong repulsive interactions, can be stabilized in a system with attraction in the presence of the modulation. Using our theoretical framework we find the plateau amplitude and shape, and characterize the correlated phase which develops in the system due to the partial gap, namely a Luttinger liquid of electronic trions. The second observation is a sharp conductance dip below a conductance of $1 \times e^2/h$, which changes its value over a wide range when tuning the system. We theorize that it is due to resonant backscattering caused by a periodic spin-orbit field. The behavior of this dip can be reliably accounted for by considering the finite length of the electronic waveguides, as well as the interactions therein. The phenomena discussed in this work exemplify the intricate interplay of strong interactions and spatial modulations, and reveal the potential for novel strongly correlated phases of matter in systems which prominently feature both.

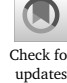

## Contents



## 1   Introduction

Low dimensional electronic systems with strong interactions present unique opportunities for the implementation and study of highly correlated quantum matter. Prominent examples are the fractional quantum Hall effect [1–3], high-$T_c$ superconductors [4], non-Fermi liquids [5–7], quantum spin liquids [8,9], and the correlated states in single and multi wall carbon nanotubes [10–13].

The two dimensional interface between the polar insulator $LaAlO_3$ and the non-polar $SrTiO_3$ features some interesting strong correlation effects, including metal-insulator transitions [14] and tunable superconductivity [15]. Recent advances in atomic force microscopy (AFM) and electron lithography have enabled confinement of these surface electrons to quasi-1D waveguides [16]. Electric conductance experiments in these heterostructures have revealed ballistic transport, and apparent strong attractive interactions between the charge carriers which lead to formation of composite few-electron particles [17,18].

Recent experiments in these quasi-1D structures have explored the introduction of an additional feature, namely spatially periodic modulation of the waveguide. This may be done in a "vertical" way, i.e., by modulating the voltage applied by the AFM tip during the patterning of the wire, which creates an effective Kronig–Penney landscape for the electrons [19]. Alternatively, one may consider "lateral" modulation, where a serpentine shape is etched at a constant AFM potential [20]. The experimental results suggest that these kind of modulations may lead to a much richer phase diagram of the electron waveguide.

Transport measurements in the vertical case have revealed a regime in which a plateau in the two-terminal conductance appears at rational fractions of the quantum of conductance [19]. Such phenomena have been previously observed in 1D constrictions [21], yet it was commonly believed strong repulsion is needed to stabilize the fractional phase [22]. In Sec. 2 we address this discrepancy, by extending the theoretical model presented in Ref. [22] to cases where the interaction has a dominant spatially periodic component. We explain the possible origin of such a form of interaction, and demonstrate that in such scenarios it is possible to measure fractional conductance in the presence of strong *attractive* interactions, provided two electronic modes have fillings approximately commensurate with one another. The presented theory is thus consistent with the total absence of such a fractional feature in the straight wires fabricated with the same technique, as we conjecture that the periodic interaction originates in the modulated patterning. Our analysis allows us to reliably recreate the plateau behavior observed in experiments, and to understand the intriguing many-body correlated phase which develops "on the plateau". We also find that another intriguing effect observed in the vertically modulated waveguides, namely an enhanced electron pairing regime, may also be accounted

for by the spatially periodic attraction.

In the laterally modulated waveguides we address a different anomaly in the transport data, namely the emergence of a conductance dip. Interestingly, this dip seems to develop and deepen continuously with change of experimental parameters, i.e., gate voltage and magnetic field [20]. In contrast to the conductance plateau in the vertical case, the conductance changes over a much wider range. Moreover, this feature is adjacent to a conductance plateau of $e^2/h$. These differences indicate that a different mechanism is at work here. In Sec. 3 we theorize this feature may be accounted for by the presence of a modulation-induced spin-orbit interaction, leading to an effective periodic potential felt by the electrons. Matching the modulation wave vector and the electron Fermi momenta leads to a resonance condition for suppressed conductivity. We show that the interplay of strong interactions in the waveguide and the short length of the conductor results in the sensitivity of the conductance dip shape and location to external parameters. Moreover, we explain how this feature can be used as a powerful probe on the strength of the interactions in the system.

## 2 Vertical modulation: fractional conductance plateau and 1-2-trion phase

As will be later shown in this Section, some of the more remarkable results in the vertically modulated waveguides can be accounted for by considering spatially modulated electron-electron interactions. It was recently established [23] that the strength of interactions, as well as their sign (repulsion or attraction) may be tuned in exactly such one-dimensional $LaAlO_3/SrTiO_3$ waveguides, by variation of the electronic density. This variation supposedly induces a kind of Lifshitz transition [23,24], modifying the orbital nature of the charge carriers, along with the effective interactions.

We conjecture that the same sort of mechanism may be at play when the "depth" of the waveguide is modulated by the varying AFM potential. As the strength and sign of interactions oscillate along the wire, the interaction becomes peaked in Fourier space, along the corresponding modulation wavevector. The consequences and significance of this modulation will become apparent in the following discussion.

### 2.1 Model

Our theoretical framework consists of a 1D system hosting two modes of spinless fermions, see Fig 1a. These modes represent the two lowest-lying electronic modes of the waveguide at a given magnetic field, as the magnetic field significantly modifies the non-interacting band structure, through both Zeeman and orbital effects [17]. The spin label of these two modes and their spatial distribution in the cross-section of the waveguide is immaterial for the purposes of this work, as long as these are two distinct modes. We consider the Hamiltonian (setting $\hbar = 1$)

$$H = \int dx \Psi_i^\dagger(x)\left(-\frac{1}{2m_i}\partial_x^2 - \mu_i\right)\Psi_i(x) + \int dx \int dy\, \rho_i(y)\mathbf{U}^{ij}(|x-y|)\rho_j(x), \quad (1)$$

where $\Psi_i(x)$ annihilates a fermion of mode $i$ at position $x$, $m_i$ and $\mu_i$ are the mass and chemical potential of the $i$-th mode, $\rho_i \equiv \Psi_i^\dagger\Psi_i$, $\mathbf{U}$ is an interaction matrix, and summation over repeated indices is implicit. Of particular interest will be the case where $\mathbf{U}$ has a contribution which is *periodically modulated in space*. This will manifest in the Fourier transform of the interaction $\mathbf{U}_q^{ij} = \int dx e^{iqx}\mathbf{U}^{ij}(x)$, which will become peaked around a specific $q^*$ corresponding to the

modulation wavevector. Notice that the form of interaction in Eq. (1) is written in a momentum conserving manner.

The low-energy physics of the model Eq. (1) may be described by linearizing the fermionic spectra near their respective Fermi momenta $k_{i,F}$, and writing the Hamiltonian in terms of right- and left-moving modes. The Hamiltonian is comprised of two contributions. The "free" part describing the linearly dispersing chiral movers,

$$\mathcal{H}_0 = i v_i \left( \psi^\dagger_{i,R} \partial_x \psi_{i,R} - \psi^\dagger_{i,L} \partial_x \psi_{i,L} \right), \tag{2}$$

where $\psi_{i,R/L}$ annihilates a right/left moving fermion of mode $i$, and $v_i$ are the Fermi velocities. The electron-electron interactions are described by

$$\mathcal{H}_{\text{int}} = g_i \rho_{i,R} \rho_{i,L} + g_\perp \left( \rho_{1,R} \rho_{2,L} + \rho_{2,R} \rho_{1,L} \right) + g_{\text{bs}} \left( \psi^\dagger_{1,R} \psi^\dagger_{2,L} \psi_{2,R} \psi_{1,L} + \text{h.c.} \right), \tag{3}$$

where $\rho_{i,r} \equiv \psi^\dagger_{i,r} \psi_{i,r}$ ($r = R, L$), and we have omitted the so-called "$g_4$ interactions" of the form $\rho_r^2$, which only renormalize the velocities later on. The different coupling coefficients may be extracted from the different momentum components of the interaction matrix,

$$g_i = \mathbf{U}^{ii}_0 - \mathbf{U}^{ii}_{2k_{i,F}}, \quad g_\perp = \mathbf{U}^{12}_0, \quad g_{\text{bs}} = \mathbf{U}^{12}_{2k_{1,F}}. \tag{4}$$

The backscattering interaction $g_{\text{bs}}$ conserves momentum (and is thus relevant at low energies) only when the Fermi momenta of the two waveguide modes are nearly identical, i.e., $k_{1,F} \approx k_{2,F}$.

We now consider a scenario where the Fermi momentum of one mode is nearly an integer multiple of the other,

$$k_{1,F} = n k_{2,F}, \tag{5}$$

facilitating a higher order backscattering term,

$$\mathcal{H}_\lambda = \lambda \psi^\dagger_{1,R} \psi_{1,L} \left( \psi^\dagger_{2,L} \psi_{2,R} \right)^n + \text{h.c.}, \tag{6}$$

*which conserves momentum* and is therefore potentially relevant. For the sake of clarity, we will focus our discussion on the case $n = 2$ (illustrated in Fig. 1b), which is the simplest possible scenario. The arguments we present here may be generalized to higher $n$ in a straightforward manner [22].

The interacting model we have presented here, captured by the effective Hamiltonian density

$$\mathcal{H} = \mathcal{H}_0 + \mathcal{H}_{\text{int}} + \mathcal{H}_\lambda, \tag{7}$$

can best be analyzed in the framework of abelian bosonization [5, 25, 26]. This is done by expressing the chiral fermionic operators in terms of new bosonic variables,

$$\psi_{i,r} \sim \frac{\eta_{i,r}}{\sqrt{2\pi\alpha}} \exp\left[ i\theta_i - ir \left( \phi_i + k_{i,F} x \right) \right], \tag{8}$$

with $r = \pm$ corresponding to $R, L$, $\alpha$ is the short-distance cutoff of our continuum model, $\eta_{i,r}$ are Klein factors ensuring fermionic commutation relations such that $\{\eta_\mu, \eta_\nu\} = 2\delta_{\mu\nu}$, and the bosonic fields obey the algebra $[\phi_i(x), \partial_x \theta_j(x')] = i\pi\delta(x - x')\delta_{i,j}$. Before writing down our bosonized model, we perform one final step: a canonical transformation on the bosons implementing a change of basis,

$$\phi_g = \frac{\phi_1 - 2\phi_2}{\sqrt{5}}, \quad \phi_f = \frac{2\phi_1 + \phi_2}{\sqrt{5}}, \tag{9}$$

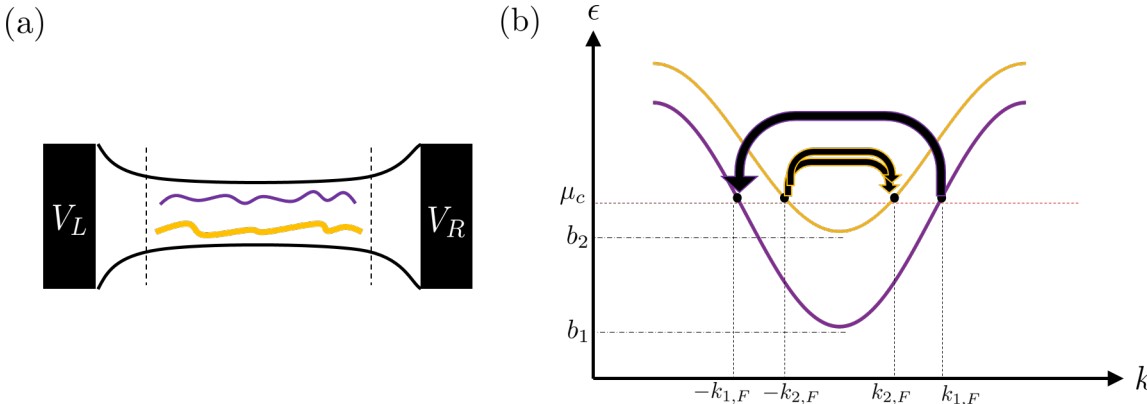

Figure 1: Schematic of the system we analyze: (a) Two non-interacting leads are attached to the strongly interacting two-mode (purple and yellow) electronic waveguide (center). The smooth broadening of the waveguide occurs on a length scale much larger than the Fermi wavelength, preventing interface reflections. (b) Qualitative dispersion of the two electronic modes in the waveguide. The dotted red line marks the Fermi level, $b_1$ and $b_2$ denote the bands bottoms and the Fermi momenta $\pm k_{1,F}, \pm k_{2,F}$ are indicated. The figure illustrates the multi-particle backscattering process we focus on: a right moving mode-1 particle (purple) backscatters off two left moving mode-2 particles (yellow). This process and its conjugate conserve momentum if $k_{1,F} = 2k_{2,F}$. This condition is satisfied when the Fermi level is at the critical chemical potential $\mu_c$.

with the same transformation for the $\theta$ operators. In this new basis, the bosonized Hamiltonian density may be written in the Luttinger liquid form,

$$
\mathcal{H} = \sum_{j=f,g} \frac{u_j}{2\pi} \left[ \frac{1}{K_j} \left( \partial_x \phi_j \right)^2 + K_j \left( \partial_x \theta_j \right)^2 \right]
$$
$$
+ \frac{1}{2\pi} \left( V_\phi \partial_x \phi_f \partial_x \phi_g + V_\theta \partial_x \theta_f \partial_x \theta_g \right) + \frac{\lambda}{4(\pi\alpha)^3} \cos\left( 2\sqrt{5}\phi_g \right). \tag{10}
$$

The explicit general form of the different parameters in the Hamiltonian (10) in terms of the different interaction strengths are given in Appendix A. Notice that we have omitted the $g_{bs}$ interaction term, as it violates momentum conservation near the range of validity of Eq. (5).

## 2.2 Fractional conductance in the strong backscattering limit

The physics we are interested in concerns the fate of the $\phi_g$ cosine term. Specifically, we begin by considering the limit $\lambda \to \infty$. We will now show that this enables us to relate the experimental signature of a fractional two-terminal conductance plateau to a gap opening in the $g$ sector.

For the sake of completeness, we briefly give here the derivation for the fractional conductance, along the lines described in Ref. [22] and its Supplementary Materials. We adiabatically attach non-interacting leads to the electronic waveguide (see Fig. 1a), and consider the scattering problem of incoming and outgoing currents in both modes. These currents are related by

$$
\begin{pmatrix} O_R \\ O_L \end{pmatrix} = \begin{pmatrix} \mathcal{T} & 1-\mathcal{T} \\ 1-\mathcal{T} & \mathcal{T} \end{pmatrix} \begin{pmatrix} I_R \\ I_L \end{pmatrix}, \tag{11}
$$

where $O_{R,L}$ and $I_{R,L}$ are chiral outgoing and incoming current vectors of length $N$, the number of modes in the waveguide, and $\mathcal{T}$ is a $N \times N$ matrix. In the case two-mode discussed here, $N = 2$ [1]. In terms of the $\phi_{1,2}$ bosonic variables, their elements are

$$I_{R,i} = \frac{e}{2\pi} \partial_t \frac{\theta_i - \phi_i}{\sqrt{2}}\big|_{x=\frac{L}{2}}, \quad I_{L,i} = \frac{e}{2\pi} \partial_t \frac{\theta_i + \phi_i}{\sqrt{2}}\big|_{x=-\frac{L}{2}}, \tag{12}$$

$$O_{R,i} = \frac{e}{2\pi} \partial_t \frac{\theta_i - \phi_i}{\sqrt{2}}\big|_{x=-\frac{L}{2}}, \quad O_{L,i} = \frac{e}{2\pi} \partial_t \frac{\theta_i + \phi_i}{\sqrt{2}}\big|_{x=\frac{L}{2}}. \tag{13}$$

In the asymptotic limit we are considering, $\lambda \to \infty$, $\phi_g$ is pinned throughout the system, and we have the boundary condition $\partial_t \phi_1 - 2\partial_t \phi_2 = 0$. Taken at opposite ends of the system, this boundary condition is equivalent to

$$\mathbf{n}_g^T \mathcal{T} = 0, \tag{14}$$

with $\mathbf{n}_g = \frac{1}{\sqrt{5}}(1, -2)^T$, defined in accordance to Eq. (9). The unobstructed propagation of the $\phi_f$ mode through the system leads to the boundary conditions $2O_{R/L,1} + O_{R/L,2} = 2I_{R/L,1} + I_{R/L,2}$, or equivalently,

$$\mathbf{n}_f^T \mathcal{T} = \mathbf{n}_f^T, \tag{15}$$

and $\mathbf{n}_f = \frac{1}{\sqrt{5}}(2, 1)^T$, which forms an orthonormal set with $\mathbf{n}_g$. The solution to Eqs. (14),(15) can be readily found to be $\mathcal{T} = 1 - \mathbf{n}_g \mathbf{n}_g^T$.

The total current flowing through the system in both modes may be expressed as $J = (1, 1) \cdot (I_R - O_L)$. Assuming incoming right movers emanate from a reservoir at potential $V$ and the left movers from a reservoir with zero potential, we set $I_R = \frac{e^2}{h} V (1, 1)^T$, and $I_L = (0, 0)^T$. The two-terminal conductance of the waveguide can then be extracted,

$$\frac{G}{e^2/h} = (1, 1)\mathcal{T}(1, 1)^T = \frac{9}{5}. \tag{16}$$

A robust conductance plateau as a function of external gate voltage at $\sim 9/5$ was experimentally observed in Ref. [19] for a wide range of magnetic fields (3-7 T). This remarkable agreement between theory and the experimental data strengthen our assertion that the observed plateaus originate in high order backscattering interactions, enabled by approximately commensurate fillings of the two modes. We further note that for the $n = 3$ case, the conductance is predicted to be $G = \frac{8}{5} \frac{e^2}{h}$. A plateau at this value of conductance, albeit much fainter than $\frac{9}{5}$ on, is also present in Ref. [19] at magnetic fields $\sim 9$ T.

The appearance of fractional conductance plateaus at a certain range of magnetic fields, as well as the possible plateau "evolution" with magnetic field, are both consistent with the well-understood role of the field in determining the band structure [17]. The magnetic field shifts the low-lying modes in energy, such that the commensurability condition Eq. (5) is made possible at a certain gate voltage. The magnetic field thus plays the role of a control parameter enabling and disabling particular many-body scattering processes.

Experimental verification of the backscattering-induced fractional conductance is possible by measuring the tunneling shot-noise "on the plateau" [22,27]. We find that for the $n = 2$ case one expects a Fano factor $e^*/e = 3/5$, whereas in the $n = 3$ case one should find $e^*/e = 2/5$ [22].

In the next section, we will explain how the varying magnetic field affects which momentum conserving backscattering interaction becomes relevant, and consequently which plateaus consequently emerge.

---

[1] The generalization to arbitrary $N$, as well as an arbitrary backscattering process, is straightforward, and appears in Ref. [22].

## 2.3 Spatially modulated interactions

As was previously discussed in Ref. [22], the relevance of the $\lambda$ perturbation, and hence the formation of the $\phi_g$ gap and fractional plateau, hinge on very strong *repulsive* interactions between the 1D electrons. However, there is strong evidence for attractive interactions in the electron waveguide devices patterned on the LaAlO$_3$/SrTiO$_3$ interface [17,18]. The key to understanding this apparent discrepancy may lie in another intriguing observation, namely that fractional conductance features were altogether absent from such straight non-modulated waveguides [17]. This hints at the possibility that the periodic modulation helps facilitate the formation of a partial gap.

To gain a qualitative understanding of how $\phi_g$ can become gapped even in the presence of attractive interactions, we examine the renormalization group (RG) flow of the Hamiltonian Eq. (10).

At each step of the RG, short-distance (or high-momentum "fast") degrees of freedom are integrated out, and the short-distance cutoff is rescaled $\alpha \to \alpha(1+d\ell)$. This generates new terms in the Hamiltonian, leading to a modification of the coupling constants [28]. Treating $\lambda, V_\phi, V_\theta$ terms as perturbations of the Luttinger liquid Hamiltonian, The second order RG equations are given by

$$\frac{d}{d\ell}\tilde{\lambda} = \left(2 - 5K_g\right)\tilde{\lambda}, \tag{17a}$$

$$\frac{d}{d\ell}K_g^{-1} = \frac{5}{4}\tilde{\lambda}^2, \tag{17b}$$

with $\ell$ the RG flow parameter, and the dimensionless coupling constant $\tilde{\lambda} \equiv \lambda/\left(2\pi^2\alpha u_g\right)$. We note that $V_{\phi/\theta}$ modify the scaling dimension of $\tilde{\lambda}$ only to second order in $V_{\phi,\theta}/u_{f,g}$, and thus introduce third-order (and higher) perturbative corrections to Eq. (17a). It is clearly evident from Eq. (17) that $K_g$ is the crucial parameter determining the fate of the $\phi_g$ sector. Neglecting its flow (which is justified if the bare $\tilde{\lambda}$ is sufficiently small), we find that the condition for a gap to open in this sector is $K_g < K_{g,c} = \frac{2}{5}$.

Let us now consider the consequence of a peaked $\mathbf{U}_q$, specifically around $q^* = 2k_{2,F}$, such that the dominant coupling coefficient is $\mathbf{U}_{2k_{2,F}}^{22}$. For simplicity, we assume all other intra-mode interaction matrix elements are of comparable strength, $\mathbf{U}_0^{11} \approx \mathbf{U}_0^{22} \approx \mathbf{U}_{2k_{1,F}}^{11} \equiv U$. Under these assumptions,

$$K_g = \sqrt{\frac{\frac{5}{2}\pi\nu + \mathbf{U}_{2k_{2,F}}^{22} + \mathbf{U}_0^{12} - U}{\frac{5}{2}\pi\nu - \mathbf{U}_{2k_{2,F}}^{22} - \mathbf{U}_0^{12} + U}}. \tag{18}$$

(See Appendix A for the full general expression.) Keeping in mind that the interactions are attractive, hence the couplings are all *negative*, one finds that with strong interactions, sufficiently dominant $\mathbf{U}_{2k_{2,F}}^{22}$ indeed greatly diminishes the value of $K_g$ and may possibly bring it below the critical $K_{g,c}$. This is a central observation of this work, which identifies the modulated interaction as an important ingredient in the fractional plateau puzzle.

## 2.4 Relation to experimental results

In the vicinity of $K_g^* = \frac{1}{5}$, which corresponds to strong interactions in the waveguide, we may supplement our asymptotic calculation Eq. (16) by an *exact* re-fermionization solution, extensively described in Ref. [22]. The point $K_g = K_g^*$ represents a generalization of the exactly solvable Luther-Emery point [29] of attractive spin-degenerate electrons in a quantum wire. We emphasize that although a Luttinger parameter much smaller than 1 usually corresponds

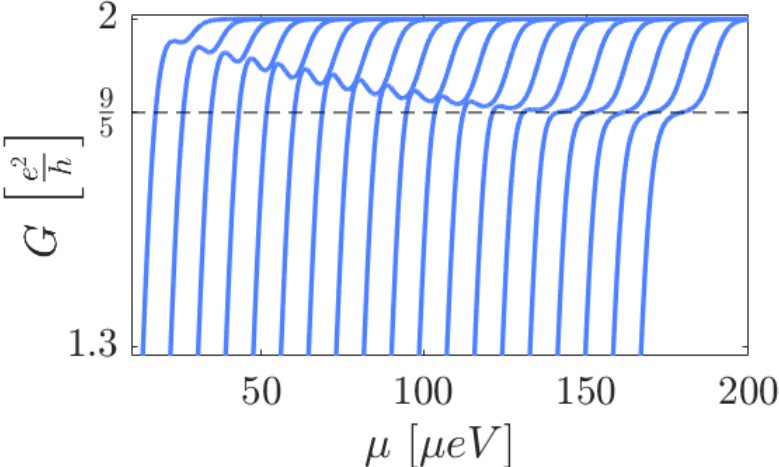

Figure 2: Two-terminal conductance in the presence of a gap in the $\phi_g$ sector, calculated from Eq. (19) and Eq. (20). The different traces correspond to an increasing gap $\Delta$ (from left to right), from $2\,\mu$eV to $20\,\mu$eV, in $1\,\mu$eV steps. Consecutive traces are shifted horizontally by $8.5\,\mu$eV for clarity. Notice that as the gap increases, the plateau approaches its asymptotic value (marked by the dashed line). We use the parameters $b_1 = 10\,\mu$eV, $b_2 = 14\,\mu$eV, $\mu_c = 26\,\mu$eV, and $T = 25$ mK for all traces.

to strong repulsive interaction, a modulated interaction may indeed lead to $K_g \ll 1$ for an *attractive* interaction as well. (See Eq. (18).)

In the limit where the length of the waveguide $L$ is sufficiently long, $L \gg \frac{\Delta}{u_g}$ with $\Delta$ the gap opened in the $\phi_g$ sector, one recovers the finite temperature linear conductance

$$G(\mu, T) = \frac{e^2}{h} \int d\epsilon \frac{\mathcal{T}(\epsilon)}{4T \cosh^2\left(\frac{\epsilon - \mu}{2T}\right)}, \tag{19}$$

with $\mu$ a global chemical potential controlled by a gate voltage, $T$ the temperature, and the transmission function, which depends on the energies of the bottom of the two bands $b_{1,2}$ (see Fig. 1), and on the critical commensurate chemical potential $\mu_c$ (see Fig. 1b),

$$\mathcal{T}(\epsilon) = \begin{cases} 0, & \epsilon < b_1 \\ 1, & b_1 \leq \epsilon < b_2 \\ 2, & b_2 \leq \epsilon < \mu_c - \frac{\Delta}{2} \\ \frac{9}{5}, & \mu_c - \frac{\Delta}{2} \leq \epsilon < \mu_c + \frac{\Delta}{2} \\ 2, & \mu_c + \frac{\Delta}{2} \leq \epsilon \end{cases} . \tag{20}$$

In Fig. 2 we show an example of the predicted conductance with different sizes of $\Delta$. A striking resemblance to the data shown in Fig. 3E of Ref. [19] is evident. Even a "pre-plateau" conductance peak feature that was seen in experiments is recreated: it is attributed to a region with transmission of 2 preceding the fractional regime, which at finite temperature does not allow the conductance to reach all the way up to its integer value. We note that the sometimes-missing plateaus at integer values of 1 and 2 in the experiment can be accounted for by an interplay between the size of the gap, the temperature and the inter-band separations. (in Fig. 2 for example, the plateau at 1, which is outside the plotted range, is smeared out for our choice of parameters.)

We may gain further qualitative insight by examining the gap $\Delta$ itself. Its size may be approximated by integrating Eq. (17a) up to $\tilde{\lambda} \sim \mathcal{O}(1)$,

$$\Delta \approx W \left( \frac{\lambda}{W} \right)^{\frac{1}{2-5K_g}}, \tag{21}$$

with $W = u_g/\alpha_0$ a typical bandwidth parameter ($\alpha_0$ is the bare short-distance cutoff). As expected, $\Delta$ becomes larger as $K_g$ is reduced. According to our theory, in the experiment the value of $q^*$ around which the interaction is peaked, remains constant. However, as the external magnetic field is modified, the energy dispersions of the two populated modes change, and the value of $k_{2,F}$ at the gate voltage corresponding to the commensurate condition (5) depends on the magnetic field. As $2k_{2,F}$ drifts further away from $q^*$, $U_{2k_{2,F}}^{22}$ becomes less dominant, and $K_g$ grows. Thus, the mechanism we present to account for the fractional plateaus in the system is entirely consistent with $\Delta$ depending on the magnetic field, and thus has remarkable agreement with the experimental variations. The same reasoning accounts for the appearance of a 9/5 plateau in one regime, and of a 8/5 plateau in another one, corresponding to $n = 2$ and $n = 3$, respectively.

We will now argue that our analysis suggests that the fractional features in the system we study, with modulated attractive interactions, are more robust as compared to the uniform repulsion case. This happens because scattering from impurities is less relevant in the former, and sometimes become irrelevant in the RG sense.

Consider the situation where $\phi_g$ is gapped and the only gapless sector is $\phi_f$. Then, the relevance of any impurity scatterer term will depend only on $K_f$ (times a numerical factor of $\mathcal{O}(1)$ determined by how the impurity impacts the two original modes). Employing an additional simplification $\mathbf{U}_0^{12} \approx U$ we may write

$$K_f = \sqrt{\frac{2\pi\nu + \frac{1}{5}\mathbf{U}_{2k_{2,F}}^{22} - U}{2\pi\nu - \frac{1}{5}\mathbf{U}_{2k_{2,F}}^{22} + U}}, \tag{22}$$

such that even for relatively dominant $\mathbf{U}_{2k_{2,F}}^{22}$ one would still expect $K_f$ to be controlled by the interaction $U$ and thus be significantly larger than in the repulsive case (remember $U < 0$). Large $K_f$ causes the impurities to be less relevant [30, 31], and the viability of the fractional conductance plateau with a value close to its asymptotic rational value rises substantially, even in a non-ideal "dirty" system. We note that the relevance of impurities in the modulated system, and hence the size of $K_f$, can be experimentally probed by introducing imperfections to the quasi-1D waveguide in its writing process. If the effect of these impurities on the conductance increases steeply with temperature, one may infer that $K_f$ is much larger than 1.

We note here that for the same reasons larger $K_f$ is expected to render proximity induced superconductivity in the residual sector much more relevant. This may possibly enable the stabilization of fractional Majorana zero modes at the edges of the waveguide [32] at more accessible temperature and proximity strength regimes.

## 2.5 Strong coupling – the 1-2-Trion phase

It is worth pointing out that the expressions we have found for the Luttinger parameters [see Eq. (18)], and their dependence on the interaction matrix elements are correct only for weak coupling, i.e., when the typical bandwidth of the two modes $W$ is sufficiently larger than the size of the elements comprising the interaction matrix $\mathbf{U}$. Furthermore, our RG arguments regarding the relevance of the multi-particle backscattering term were also perturbative. For very strong interactions, the functional dependence of, e.g., the size of the gap (or its existence), may vary.

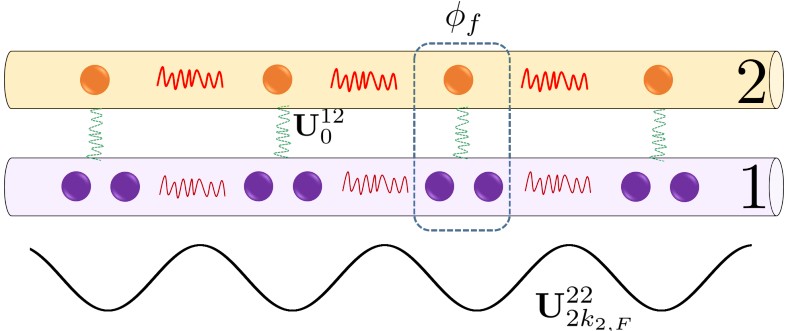

Figure 3: Strong coupling limit of the two commensurate waveguide modes, cf. Fig.1. Attractive interactions with wave-vector $2k_{2,F}$ (solid black line at the bottom) corresponding to the less populated mode (mode 2), tend to induce a charge density wave commensurate with that wave vector in each mode. The inter-mode attraction (wiggly green lines) then "locks" the phases of the two density waves together. This locking corresponds to pinning $\phi_g$, whereas a composite 1-2-trion $\phi_f$ can propagate along the waveguide.

We claim, however, that qualitatively one should reach the same conclusions in the strong interaction limit. To understand why, consider the limit of negligible electron hopping and only interactions of the kind we have discussed, see Fig. 3. If the intra-mode attractive interaction has a dominant component modulated with a spatial frequency matching the density of the less populated mode $\mathbf{U}_{2k_{2,F}}^{22}$, the most energetically favored state would be a charge density wave with the corresponding wave vector, which will maximize the attraction. Then, the subdominant inter-mode attraction will tend to "glue" a mode-2 particle to two mode-1 particles. This corresponds to the free $\phi_f$ mode left in the waveguide in the language of our previous discussion. According to Eq. (9) it is composed of 2 bosonic modes from mode-1 and one from mode-2. Notice that in the expression for $K_g$, Eq. (18), from which we determine the fate of $\phi_g$, the weak coupling dependence on the coupling constants $\mathbf{U}_{2k_{2,F}}^{22}, \mathbf{U}_0^{12}$ reflects in essence the strong coupling heuristic description, as large attractive (with negative amplitudes) interaction tend to make $K_g$ small and pin $\phi_g$. We note that the above argument is not specific to $n = 2$, and generally holds, with proper modifications, for arbitrary $n$.

We now comment on the state of the electrons in the waveguide "on the plateau", i.e., deep in the $\phi_g$ gapped phase. Let us consider the operator

$$\Psi_{\text{1-2-trion}}(x) \equiv \Psi_1(x)\Psi_1(x+\alpha)\Psi_2(x)$$
$$\propto e^{-i\sqrt{5}\theta_f}\cos\left(\frac{\phi_f}{\sqrt{5}} + k_{2,F}x\right), \tag{23}$$

which creates a three-particle fermionic excitation, as in Fig. 3. The offset by $\alpha$ in the second annihilation operator is crucial in order to create a local pair due to the fermionic nature of $\Psi_1$. This is the lowest order operator one can construct that does not contain the dual variable $\theta_g$, which strongly oscillates in the fractional phase leading to exponentially decaying correlation functions of all operators containing it. We may then examine trion-density-density and trion-pair correlations,

$$\langle\rho_{\text{1-2-trion}}(x)\rho_{\text{1-2-trion}}(0)\rangle \propto \cos\left(2k_{2,F}x\right)x^{-\frac{2K_f}{5}}, \tag{24}$$

$$\left\langle\Delta_{\text{1-2-trion}}^\dagger(x)\Delta_{\text{1-2-trion}}(0)\right\rangle \propto x^{-\frac{10}{K_f}}, \tag{25}$$

respectively, with $\rho_{\text{1-2-trion}}(x) = \Psi^{\dagger}_{\text{1-2-trion}}(x)\Psi_{\text{1-2-trion}}(x)$, and $\Delta_{\text{1-2-trion}}(x) = \Psi_{\text{1-2-trion}}(x)\Psi_{\text{1-2-trion}}(x+\alpha)$. The value of $K_f$ determines which of these two will be the dominant order in the gapped system: for $K_f < 5$ charge density wave order of composite trions will dominate, whereas trion pairing will have the leading susceptibility if $K_f > 5$.

## 2.6 Enhanced pairing

Another remarkable phenomenon observed in the vertically modulated waveguides is an extended regime where electrons form bound pairs [19]. We now demonstrate that this experimental signature is entirely consistent with the conjectured periodic attractive interaction.

We examine the Hamiltonian density $\mathcal{H}_0 + \mathcal{H}_{\text{int}}$ [Eqs. (2) and (3)] around the commensurability point $k_{1,F} = k_{2,F} \equiv k_F$. Thus, $g_{\text{bs}}$ is a relevant perturbation, whereas $\mathcal{H}_\lambda$ is not (and thus omitted). For simplicity, we assume the difference between the various intra- and inter-mode interactions are negligible, i.e., $\mathbf{U}_0^{11} \approx \mathbf{U}_0^{22} \approx \mathbf{U}_0^{12}$, and $\mathbf{U}_{2k_F}^{11} \approx \mathbf{U}_{2k_F}^{22} \approx \mathbf{U}_{2k_F}^{12} \equiv U_{2k_F}$. Simplifying further, $v_1 \approx v_2 \equiv v$, we find that the bosonized Hamiltonian density can be written as

$$\mathcal{H}_{\text{pair}} = \sum_{\eta=+,-} \frac{u_\eta}{2\pi} \left[ \frac{1}{K_\eta} \left( \partial_x \phi_\eta \right)^2 + K_\eta \left( \partial_x \theta_\eta \right)^2 \right] + \frac{U_{2k_F}}{2(\pi\alpha)^2} \cos\left( \sqrt{8}\phi_- \right), \tag{26}$$

with $\phi_\pm = \frac{\phi_1 \pm \phi_2}{\sqrt{2}}$, and an identical transformation is applied for $\theta_\pm$. (The parameters $u_+$, $u_-$, and $K_+$ are not important for our discussion and are given in Appendix A.) Crucially, we find [25]

$$K_- = \sqrt{\frac{1 + U_{2k_F}/(2\pi v)}{1 - U_{2k_F}/(2\pi v)}}, \tag{27}$$

which is smaller than 1 for attractive interactions, making the cosine term relevant in the RG sense. When this term flows to strong coupling, $\phi_-$ gets pinned, and only pairs with one electron from each mode (corresponding to the $\phi_+$ channel) remain gapless. Integrating the RG flow up to strong coupling, we can estimate the pairing gap [similarly to Eq. (21)],

$$\Delta_{\text{pair}} \approx W \left( \frac{U_{2k_F}}{W} \right)^{\frac{1}{2-2K_-}}. \tag{28}$$

We now consider the impact of modulated attractions, such that $\mathbf{U}_q$ peaks near $q = 2k_F$. Clearly, this leads to a significant enhancement of $\left| U_{2k_F} \right|$ as compared to the more generic short-range or power-law decaying interactions. This in turn increases the size of the gap $\Delta_{\text{pair}}$, as it makes $K_-$ smaller and the ratio $U_{2k_F}/W$ larger. An enhancement of the pairing region as compared to the non-modulated case (cf. Ref. [17]) can thus be attributed to the modulated attractive interaction. This conjectured form of interaction can thus account for *both* of the most prominent features observed in the vertically modulated waveguides.

## 3 Lateral modulation: Gap opening and Reduction of conductance

The effect of lateral modulation of the electron waveguide may be captured by an alternating electric field in the lateral direction with wave vector $Q$, $\mathbf{E} = E \cos(Qx)\hat{y}$, felt by the electrons having momenta $\mathbf{k} = k\hat{x}$. An effective modulated Rashba spin-orbit field $\boldsymbol{\alpha}$ in the out-of-plane direction is thus expected, as $\boldsymbol{\alpha} \propto \mathbf{k} \times \mathbf{E} = kE \cos(Qx)\hat{z}$. In this Section, we explore the

consequences of a modulated spin-orbit interaction in the high (out-of-plane) magnetic field regime.

Focusing on the lowest-lying spinfull mode in the waveguide, we describe it by an Hamiltonian $H = \int dx \Psi^\dagger \left[ \mathcal{H}_0 + \mathcal{H}_Q \right] \Psi$, where $\Psi = \left( \psi_\uparrow, \psi_\downarrow \right)^T$ is a spinor of electron annihilation operators, and $\mathcal{H}_0$ describes the system without modulation,

$$\mathcal{H}_0 = -\frac{\partial_x^2}{2m} - \epsilon_0 + V_Z \sigma_z - \alpha_0 i \partial_x \sigma_z, \tag{29}$$

with $m$ being the electron mass, $\epsilon_0$ is the Fermi energy in the absence of Zeeman splitting and spin-orbit coupling, $V_Z$ is the Zeeman energy, $\alpha_0$ is the non-modulated component of the spin-orbit interaction, and $\sigma_z$ is a Pauli operator. The modulated spin-orbit interaction is described by

$$\mathcal{H}_Q = \alpha_Q \left\{ -i \partial_x, \cos\left( Qx \right) \right\} \sigma_z, \tag{30}$$

with $\alpha_Q$ the strength of the modulated spin-orbit coupling, and the anti-commutator ensures the hermiticity of the Hamiltonian. Such a form of spin-orbit interaction was considered in Ref. [33], where a metal-insulator transition was studied. Considering here the regime $V_Z \gg \epsilon_0 + \frac{m\alpha_0^2}{2}$ (taking $V_Z$ positive without loss of generality), we may limit our discussion to the low-energy $\sigma_z = -1$ sector, as depicted in Fig. 4a. Linearizing the spectrum of $\mathcal{H}_0$ around $k = \pm k_F + k_{SO}$, with $k_F = \sqrt{2m\left( \epsilon_0 + V_Z \right) + m^2 \alpha_0^2}$, and $k_{SO} = m\alpha_0$, we expand

$$\psi_\downarrow \approx e^{i(k_F + k_{SO})x} \psi_R + e^{-i(k_F - k_{SO})x} \psi_L, \tag{31}$$

with $\psi_{L/R}$ being chiral fermionic operators. The total Hamiltonian density projected to the $\sigma_z = -1$ sector may then be expressed as

$$\mathcal{H} = iv\left( \psi_R^\dagger \partial_x \psi_R - \psi_L^\dagger \partial_x \psi_L \right) + \alpha_Q k_{SO} \left( \psi_R^\dagger \psi_L e^{-i(2k_F - Q)x} + \text{h.c.} \right), \tag{32}$$

where we have omitted rapidly oscillating terms, and $v = k_F / m$. It is apparent from Eq. (32) that in our parameter regime of interest the system is equivalent to one of spinless fermions subjected to a spatially periodic potential. This periodic potential is due to the "dc" and periodic components of the spin-orbit interaction conspiring together. Performing a unitary transformation $\psi_{R/L} \to \psi_{R/L} \exp\left( \pm i \frac{2k_F - Q}{2} x \right)$, Eq. (32) becomes

$$\tilde{\mathcal{H}} = iv\left( \psi_R^\dagger \partial_x \psi_R - \psi_L^\dagger \partial_x \psi_L \right) - \tilde{\mu}\left( \psi_R^\dagger \psi_R + \psi_L^\dagger \psi_L \right) + \Delta_Q \left( \psi_L^\dagger \psi_R + \psi_R^\dagger \psi_L \right), \tag{33}$$

where $\tilde{\mu} = \frac{v}{2}\left( 2k_F - Q \right)$, and $\Delta_Q = \alpha_Q k_{SO}$.

## 3.1 Conductance

When the spin-orbit spatial frequency $Q$ exactly matches $2k_F$, so that $\tilde{\mu} = 0$, the impact of $\Delta_Q$ is maximal, as is illustrated in Fig. 4a. The transmission coefficient for a (non interacting) system of length $L$ at the center of the gap is given by, (see Appendix B):

$$\mathcal{T}_L = \frac{1}{\cosh^2\left( \frac{\Delta_Q}{v} L \right)}, \tag{34}$$

and thus we can use Landauer's two-terminal conductance formula in Eq. (19) to calculate the conductance with the approximated transmission function

$$\mathcal{T}(\epsilon) = \begin{cases} 0, & \epsilon < 0 \\ 1, & 0 \leq \epsilon < \mu_Q - \Delta_Q \\ \mathcal{T}_L, & \mu_Q - \Delta_Q \leq \epsilon < \mu_Q + \Delta_Q \\ 1, & \mu_Q + \Delta_Q \leq \epsilon \end{cases}, \tag{35}$$

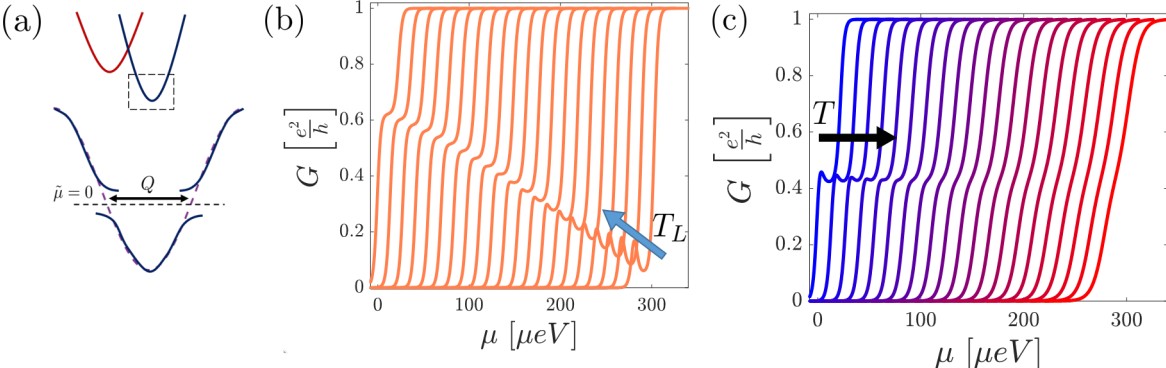

Figure 4: (a) Top: Schematic dispersion for $\mathcal{H}_0$, Eq. (29), with uniform spin orbit coupling and Zeeman field. Different colors represent opposite out-of-plane spin projections, and the vertical offset is due to an out of plane Zeeman magnetic field term. Bottom: Zoom-in on the dashed square of the left panel presenting a schematic dispersion of a 1D single spin fermion without (dashed line) and with (solid line) a periodic potential, originating in the spatially modulated spin-orbit interaction, Eq. (30). When $Q$, the lateral modulation wave vector, matches $2k_F$, $\tilde{\mu} = 0$, Eq. (33). Then the Fermi level (dashed black line) is within the induced gap. (b) Two-terminal conductance of a mode subjected to a spatially periodic potential. We use Eq. (19) and Eqs. (34)–(35) to calculate the conductance. The different traces correspond to different values of $T_L = v/L$ (from left to right) between 30 $\mu$eV and 10 $\mu$eV in steps of 1 $\mu$eV. (We expect that experimentally the application of magnetic field will affect the velocity of the mode, $v$, in the wire as discussed in Ref. [20].) For clarity, we mark the direction of increasing $T_L$ by an arrow, and consecutive traces are shifted horizontally by 14 $\mu$eV. We use the parameters $T = 25$ mK, $\Delta_Q = 10k_B T$, and $\mu_Q = 12$ $\mu$eV for all traces. (c) Similar to (b), but now varying the temperature between 2–6 $\mu$eV in steps of 0.2 $\mu$eV, and keeping $T_L = 20$ $\mu$eV constant. For clarity, we mark the direction of increasing $T$, and consecutive traces are shifted horizontally by 14 $\mu$eV. Here we use $\Delta_Q = 20$ $\mu$eV and $\mu_Q = 12$ $\mu$eV for all traces.

with $\mu_Q = \frac{Qv}{2}$. [Notice that the chemical potential used in Eq. (19) is defined with respect to the bottom of the linearized band in Eq. (33). The Fermi level in this model thus coincides with $\mu = vk_F$, and when it is equal to $\mu_Q$ one finds $\tilde{\mu} = 0$, and the backscattering is resonant.] In Eq. (35) we have simplified the transmission function, such that the transmission within the gap is approximately constant, and outside the gap it is unity. This simplified form is sufficient to capture the experimental features. The accurate transmission coefficient of the effective scattering problem may be found in Appendix B.

Calculating the conductance, with temperature of 25 mK $\approx$ 2.15 $\mu$eV that is comparable to the experimental temperature, $\Delta_Q$ ten times larger, and $T_L \equiv v/L$ that varies between about 15 to 5 $T$, we obtain the conductance depicted in Fig. 4b. A "shoulder" at the conductance around $0.6e^2/h$ for large $T_L$ develops to a pronounced dip for small $T_L$. Similar behavior is observed in the experiment, (see Fig. 2 of Ref. [20]) when the out of plane magnetic field is varied. We expect that the magnetic field will affect the velocity of the modes in the wire as can be ascertained from our expression for $k_F$, and hence $T_L$ is expected to vary, as we plot in Fig. 4b.

### 3.2 Interactions

The discussion of the lateral modulation so far does not include a key ingredient of the experimental system, strong electron-electron interactions. We shall account for this by using, similar to the vertical case, the bosonization technique. At $\tilde{\mu} = 0$, the bosonized Hamiltonian of the system in question, connected to infinite non-interacting leads at both its ends, may be written in the form

$$\mathcal{H} = \frac{\tilde{v}}{2\pi}\left[\frac{1}{K(x)}(\partial_x \phi)^2 + K(x)(\partial_x \theta)^2\right] + w(x)\frac{\Delta_Q}{2\pi\alpha}\cos(2\phi), \tag{36}$$

where $w(x)$ is unity within the region $0 < x < L$ and zero outside of it, and $K(x) = K$ within that same region and $K(x) = 1$ in the leads. The effects of the interaction are captured by the modification $v \rightarrow \tilde{v}$, and by $K$, and for simplicity we neglect the effect of different Fermi velocities in different regions of the system. In contrast to our discussion in Sec. 2, in the laterally modulated case we do not assume a modulated interaction, and thus $K > 1$ corresponds to attractive interactions and $K < 1$ for repulsive interactions, as usual.

The lowest order RG equation for the flow of $\Delta_Q$ is

$$\frac{d}{d\ell}\Delta_Q = (2 - K)\Delta_Q, \tag{37}$$

showing that the periodic perturbation may be relevant even for moderately strong attractive interactions, as long as $K < 2$. We consider the regime in which $T \ll T_L \lesssim \Delta_Q$, such that the RG flow is cut off by the length scale of the system. Thus, at energy scales below $T_L$ we are left with the Hamiltonian of a simple backscattering impurity center embedded in a interaction-free Luttinger liquid (the leads connected to the system). The strength of this effective impurity as compared to $\Delta_Q$ depends on the nature of interactions in the system. Since the scaling of the gap in such a regime goes as $\Delta_Q = \Delta_Q^0 \left(\frac{L}{\alpha_0}\right)^{1-K}$, with $\Delta_Q^0$ the bare gap value, the transmission coefficient from Eq. (34) may be replaced by the approximation (which is valid in the vicinity of $K \approx 1$)

$$\mathcal{T}_L^* = \frac{1}{\cosh^2\left[\frac{\Delta_Q^0}{W}\left(\frac{L}{\alpha_0}\right)^{2-K}\right]}, \tag{38}$$

and once again $W = \tilde{v}/\alpha_0$ is the bandwidth parameter. By measuring the conductance of identical systems with varying length, Eq. (38) provides a probe on the interaction strength in the modulated waveguide, as well as its nature (attraction or repulsion).

We finally comment on the effect of temperature in such a regime. As long as the system remains in the regime $T < T_L$, a change of temperature will have a negligible impact on the transmission coefficient, as the model still reduces to an effective impurity backscattering center problem in a non-interacting system (i.e., the infinite leads). Thus, the *value* of the conductance dip around $\tilde{\mu} = 0$ should not change with temperature, yet its shape would be blurred (and eventually vanish) when the system is heated up. This is precisely the trend observed in Fig. 3 of Ref. [20], and is recreated with sensible parameters in our Fig. 4c. This is strong evidence supporting our conclusions regarding the parameter regime, as well as the origin of the finite transmission plateau at low densities.

## 4 Conclusions

Electron waveguides created on $LaAlO_3/SrTiO_3$ interfaces have proven themselves in recent years to be new and exciting platforms to study highly correlated electrons physics. The experiments addressed in this work, Refs. [19,20], explored the effect of waveguide modulation

on the electron transport. Two novel features were found for the two different kinds of modulation.

For the vertical case, where the "writing" potential oscillated along the wire, plateaus in the two-terminal conductance as a function of gate voltage appeared at fractional values of the quantum of conductance. The appearance of these plateaus depended on the magnetic field as well as the fillings of the modes. With lateral modulation, creating a serpentine-like trajectory for the electrons, an intriguing conductance dip emerged in the supposedly singly-occupied-mode regime. This dip appeared to vary its value, and to some extent its shape, when the external magnetic field was swept.

In this manuscript, we have presented theoretical frameworks which can account for these unusual transport phenomena. We have argued that the experimental data for the vertically modulated waveguides is consistent with the existence of two strongly interacting electronic modes, whose filling is commensurate with one another. An asymptotic theoretical analysis of the conductance for 2:1 filling ratio yields a plateau with conductance of $9/5\,e^2/h$ for a certain range of magnetic filed. Similarly a 3:1 ratio yields conductance of $8/5\,e^2/h$.

Remarkably, these two filling scenarios were previously predicted to be the most susceptible to the opening of a partial gap in the system, and thus to stabilization of fractional conductance signatures [22]. The shapes of the conductance plateaus were calculated using a re-fermionization technique in a strongly coupled regime (akin to a generalized Luther-Emery point [29]), and were found to bare qualitative resemblance to the reported experiments.

In the current work we have further argued that the spatial modulation is indispensable to the stabilization of the high-order backscattering gap in the presence of attractive interactions. We conjecture that the main role of the modulation is in making the interaction itself oscillate and peak at a specific wavevector $q^*$, through the mechanism discussed in Refs. [23, 24]. We have shown here that such an interaction indeed supports the formation of a gap leading to fractional plateaus, both in the weak-coupling RG sense, and in the strong coupling picture. We have demonstrated that the second remarkable feature observed in these wave guides, an enhancement of the electron pairing, may also be explained by the the same modulated inter-action. This lends credence to our claim that vertical modulation of the electronic waveguide can lead to a periodic interaction felt by the electrons.

The appearance of two-terminal conductance plateaus at rational fractions of the quantum of conductance $e^2/h$ with the introduction of periodic modulation to the system has profound implications. We have demonstrated that in such a scenario, contrary to previous studies concerning this fractional phenomenon, the partial gap due to strong interactions may be stabilized by electron-electron *attraction*. This suggests that the fractional conductance anomaly is perhaps more ubiquitous than it is currently believed to be, and may be realized at certain parameter regimes in other experimental platforms.

We speculate that the attractive nature of the interactions is responsible for the relative robustness of the plateaus as compared to, e.g., the plateaus observed in Ref. [21], where the experiments were performed in GaAs based split-gate quantum wires with repulsive interactions. The attraction would generically make the residual impurity back scattering, which are expected to deteriorate the transport in the repulsive scenario, less relevant. Thus, one would expect the values of the conductance plateaus to be much closer to their asymptotic values calculated by the method of Sec. 2.2.

As mentioned earlier, the conductance dip at certain fillings of the laterally modulated waveguide is qualitatively distinct from the plateau observed with vertical modulation, suggesting the two have different origins. We attribute it to an effective periodic potential felt by the propagating electrons, presumably originating in a modulation-induced spatially-periodic spin-orbit interaction. When this spin-orbit potential provides the correct momentum for a single-particle backscattering event, i.e., when $Q = 2k_F$, the electronic mode develops a gap.

For long enough waveguides, this would lead to a total suppression of the two-terminal conductance at low temperatures. However, as we explain, the experimental data suggests that the finite conductance found on this resonance is due to the finite length of the system. The observed results are consistent with the energy scale $T_L$ (which is inversely proportional to the waveguide length) and the gap energy being of comparable sizes, while the temperature is much smaller than both.

Interactions play an important role in the lateral modulation as well. They tend to renormalize the size of the gap and make it larger for repulsive interactions and moderately attractive ones, or diminish it in the case of strong enough attraction. The continuous shift of the conductance dip value as the magnetic field is swept in the experiment with lateral modulation can thus be attributed also to a change of the effective interactions, which by affecting the renormalized Fermi velocity or the gap, alter the ratio $\Delta_Q/T_L$ (defined in Sec. 3.1 and Fig. 4). Furthermore, assuming the relevant experimental regime corresponds to the gap renormalization being cut off by the finite system length $L$, varying $L$ while measuring the change in the conductance would allow one to ascertain the strength of the interaction and possibly verify its attractive nature.

As we conjecture in the beginning of Sec. 3, the transport may indicate the presence of modulated spin-orbit interactions. While these experiments [20] were conducted at high field, the modulated spin orbit coupling may lead to interesting phenomena in the absence of magnetic filed. For example, tuning the modulation wavelength and shape may enable designing high quality spin transistors with no ferromagnetic reservoirs [34].

## Acknowledgements

We thank Jeremy Levy for fruitful discussions of his experimental data.

**Funding information**   This work was partially supported by the European Union's Horizon 2020 research and innovation programme (Grant Agreement LEGOTOP No. 788715), the DFG (CRC/Transregio 183, EI 519/7-1), and the Israel Science Foundation (ISF) and by a grant from the Binational Science Foundation (BSF) and the National Science Foundation (NSF).

## A   Bosonized Hamiltonian parameters

In Sec. 2 we discuss the role of vertical modulation and use a bosonized formulation of the model, see Eq. (10). For the sake of completeness, we bring here the general form of the parameters that are used in it, in terms of the fermionic velocities and interactions appearing in Eqs. (2)−(3). Assuming for simplicity $v_1 = v_2 \equiv v$, we find

$$u_g = v\sqrt{1 - \left(\frac{g_1 + 4g_2 - 4g_\perp}{10\pi v}\right)^2}, \quad u_f = v\sqrt{1 - \left(\frac{4g_1 + g_2 + 4g_\perp}{10\pi v}\right)^2}, \tag{39}$$

$$K_g = \sqrt{\frac{10\pi v - g_1 - 4g_2 + 4g_\perp}{10\pi v + g_1 + 4g_2 - 4g_\perp}}, \quad K_f = \sqrt{\frac{10\pi v - 4g_1 - g_2 - 4g_\perp}{10\pi v + 4g_1 + g_2 + 4g_\perp}}, \tag{40}$$

$$V_{\phi/\theta} = \mp\frac{2}{5\pi}\left(g_2 - g_1 + \frac{3}{2}g_\perp\right). \tag{41}$$

The connection between the coupling constants and the interaction matrix $\mathbf{U}$ is given in Eq. (4). In the main text we use the simplification $\mathbf{U}_0^{11} \approx \mathbf{U}_0^{22} \approx \mathbf{U}_{2k_{1,F}}^{11} \equiv U$ for $K_g$ in Eq. (18),

and the additional assumption $g_\perp \approx U$ when we discussed the role of $K_f$ in Sec. 2.4. We note that under the same assumptions, one find the expression $V_{\phi/\theta} = \mp\frac{1}{\pi}\left(U - \frac{2}{5}\mathbf{U}^{22}_{2k_{2,F}}\right)$. As discussed in Sec. 2.3, these cross interactions affect the scaling dimension of the $\phi_g$ sector, yet in a quantitatively modest manner.

In Sec. 2.6 we discuss a modified Hamiltonian, valid around $k_{1,F} = k_{2,F}$, see Eq. (26). The expressions for the parameters that go into it may be expressed as

$$u_+ = v\sqrt{1 - \left(\frac{U_{2k_F} - 2\mathbf{U}_0}{2\pi v}\right)^2}, \quad u_- = v\sqrt{1 - \left(\frac{U_{2k_F}}{2\pi v}\right)^2}, \tag{42}$$

$$K_+ = \sqrt{\frac{1 + U_{2k_F}/(2\pi v) - 2\mathbf{U}_0/(2\pi v)}{1 - U_{2k_F}/(2\pi v) + 2\mathbf{U}_0/(2\pi v)}}, \quad K_- = \sqrt{\frac{1 + U_{2k_F}/(2\pi v)}{1 - U_{2k_F}/(2\pi v)}}, \tag{43}$$

where we have used the simplifications $v_1 \approx v_2 \equiv v$, $\mathbf{U}^{11}_0 \approx \mathbf{U}^{22}_0 \approx \mathbf{U}^{12}_0 \equiv \mathbf{U}_0$, and $\mathbf{U}^{11}_{2k_F} \approx \mathbf{U}^{22}_{2k_F} \approx \mathbf{U}^{12}_{2k_F} \equiv U_{2k_F}$.

## B Solving the scattering problem

Here we solve the scattering problem we discuss in Sec. 3. Let us rewrite Eq. (33) in a more convenient form,

$$H = \int dx\, \Psi^\dagger \mathcal{H}_{\text{scat}} \Psi, \quad \text{with} \quad \mathcal{H}_{\text{scat}} = iv\partial_x\sigma_z - \tilde{\mu} + \Delta_Q\sigma_x. \tag{44}$$

Here $\sigma_i$ are Pauli matrices and $\Psi = (\psi_R, \psi_L)^T$. We solve the Schrödinger equation $\mathcal{H}_{\text{scat}}\Psi = E\Psi$ using the ansatz $\Psi(x) = \exp[Fx/v]\Psi(0)$, which is justified for a translation-invariant Hamiltonian. One can readily find:

$$F = \Delta_Q\sigma_y - i(E + \tilde{\mu})\sigma_z. \tag{45}$$

To solve the scattering problem we set the boundary conditions $\Psi(0) = (1, r)^T$, $\Psi(L) = (t, 0)^T$, and find the transmission coefficient as $\mathcal{T} = |t|^2$. Overall, using $T_L \equiv v/L$, we find

$$t = \left(\cosh\sqrt{\left(\frac{\Delta_Q}{T_L}\right)^2 - \left(\frac{E + \tilde{\mu}}{T_L}\right)^2} + i\frac{E + \tilde{\mu}}{\sqrt{(\Delta_Q)^2 - (E + \tilde{\mu})^2}}\sinh\sqrt{\left(\frac{\Delta_Q}{T_L}\right)^2 - \left(\frac{E + \tilde{\mu}}{T_L}\right)^2}\right)^{-1}, \tag{46}$$

from which we recover Eq. (34) for $E = \tilde{\mu} = 0$.

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
