# Peer review of "Modulation induced transport signatures in correlatedelectron waveguides"

_SciPost Physics, doi:SciPost Phys. 9, 051 (2020)_

## Round 1 · Referee Report · Anonymous (Referee 4) · 2020-8-26

Report

The authors have addressed all referees' concerns and suggestions in an impressively careful and constructive way. The adopted changes make the paper even more outstanding.

  • validity: -
  • significance: -
  • originality: -
  • clarity: -
  • formatting: -
  • grammar: -

Author:  Gal Shavit  on 2020-09-08  [id 948]

(in reply to Report 1 on 2020-08-26)

We would like to thank the referee for reading the revised manuscript, reviewing our comments and corrections and describing the paper as outstanding.

---

## Round 1 · Referee Report · Anonymous (Referee 5) · 2020-8-27

Report

I much appreciate the improvements the authors have made to the paper and their answers to my questions. A few final issues remain to be cleared up:

p. 14 In the revision (and their reply) the authors write:
"Notice that at μ=μQ, ˜μ0, which gives the resonance condition"
I cannot follow why ˜μ must vanish: starting from the relations that I can extract from the paper on p. 12 (I eliminate v=kF/m),
k2F=2m(μ+VZ)+k2SO
and
μQ=kFQ/2,
it is not obvious how this gives the condition Q=2kF to nullify
˜μ=kF(2kFQ)/2=0 as defined on p. 12.
Please clarify which clue I am missing.

p. 13, after Eq. (33)
The definition of ΔQ seems to have changed. If it is now correct (it seems so), then
ΔQ coincides with VQ which appears in discussion after Eq. (36) alongside ΔQ which is very confusing. Please replace throughout Sec. 3.2 VQΔQ or clarify the use of different symbols for the same thing.

p. 12 bottom:
The authors have indicated that they corrected a typo in the unitary transformation. It seems that the phrase "and once more neglecting spatially oscillating terms" needs to be removed: Unless I misunderstood, Eq. (32) seems to imply Eq. (33) by working out the corrected unitary transformation.
Please clarify.

p. 4: Equation (4) seems to has changed (not indicated by the authors) but leaves me confused:
I understood gbs is just a number ("coupling coefficient"), how can there be there be a δk1F,k2F on the right hand side?
Please clarify (4) or adjust Eq. (3).
* * *
Minor typographic issues/suggestions that I noted:

Abstract, last line: "s" missing
"of matter in systemS which prominently feature both."

p. 3: ".. we explain how can this feature be used"
-> " we explain how this feature can be used"

p. 7 : Skip the paragraph break at (dangling remark)
"..spin-degenerate electrons in a quantum wire. We emphasize that .."

Fig. 2 on p. 7 is discussed only on p. 9.
Perhaps move the figure to p. 8?

p. 14: Skip the paragraph break at
..regions of the system. In contrast to our discussion.."

Requested changes

see report

  • validity: high
  • significance: high
  • originality: top
  • clarity: high
  • formatting: excellent
  • grammar: perfect

Author:  Gal Shavit  on 2020-09-08  [id 950]

(in reply to Report 2 on 2020-08-27)

We would like to thank the referee for carefully reading the revised manuscript and reviewing our comments and corrections.
We address the issues he raise below.

1)
Regarding the comment about μ and ˜μ in page 14:
The resonance condition for the backscattering is given by vkF=μQ. In a linearized Dirac Hamiltonian, e.g., Eq. (33), the Fermi energy, or the chemical potential is measured relatively to the Dirac crossing point is just μ=vkF, which is why we claim that when μ=μQ the backscattering is resonant.
However, the referee's point seems to be that this μ should not be the same as the one in Eq. (29) (which we have replaced by ϵ0 in the revised manuscript). This is a valid point. Although ϵ0 determines μ (via the definition of kF), it is not equal to it.

To correct this point, and avoid using confusing notation, we have changed μϵ0 in Eq. (29).
Additionally, we clarify the point of chemical potential being defined with respect to the linear model below Eq. (35).

2)
Regarding the redundancy between ΔQ and VQ.
Indeed, as the referee points out, these two coincide.
Thus, we replace VQ by ΔQ everywhere in the amended manuscript.

3)
Regarding the unitary transformation at the bottom of page 12.
The sentence the referee refers to should indeed be removed, and is removed in the revised manuscript.

4)
Regarding gbs in Eq. (4).
The meaning of the Kronecker delta previously used in Eq. (4) is to clarify that gbs plays a role only when the two Fermi momenta are approximately the same. Otherwise, the corresponding term will oscillate rapidly along the waveguide.
For the sake of clarity, we omit the Kronecker delta in Eq. (4), and clarify this condition following Eq. (4).

The other typos and suggestions of the referee have been fully implemented in the revised manuscript.

---

## Round 1 · Author Response

Dear Editor,
We would like to thank all three referees for their thoughtful and careful reading of the manuscript. We are glad they all found our work interesting, original, and appropriate for publication in SciPost. The comments provided by the referees mostly concern the presentation and clarifications needed to make the manuscript more accessible to the readers. We have fully implemented their comments and suggestions in the revised manuscript, which is now improved as a consequence. We have detailed the changes made to the manuscript in the response to each referee below. With this, we trust the manuscript is ready for publication in SciPost.

Sincerely yours,
Gal Shavit and Yuval Oreg.

---

## Round 1 · List of Changes

The changes made in the revised manuscript detailed in the responses to the referees.
Additionally, we have changed the formatting to conform with the SciPost format.

---

## Round 2 · Author Response

Dear Editor,
We would like to once again thank the referees for reviewing the revised manuscript. We have made some minor modifications to the manuscript following the recent questions and suggestions of Referee 2, for which we are grateful.
With this, we believe the manuscript is ready for publication.

Sincerely yours,
Gal Shavit and Yuval Oreg.

---

## Round 2 · List of Changes

The changes made in the revised manuscript detailed in the responses to the referees.

---

## Editorial Decision

published